# Interface Effects on He Ion Irradiation in Nanostructured Materials

**DOI:** 10.3390/ma12162639

**Published:** 2019-08-19

**Authors:** Wenfan Yang, Jingyu Pang, Shijian Zheng, Jian Wang, Xinghang Zhang, Xiuliang Ma

**Affiliations:** 1Shenyang National Laboratory for Materials Science, Institute of Metal Research, Chinese Academy of Sciences, 72 Wenhua Road, Shenyang 110016, China; 2School of Material Science and Engineering, University of Science and Technology of China, Hefei 230026, China; 3Tianjin Key Laboratory of Materials Laminating Fabrication and Interface Control Technology, Hebei University of Technology, Tianjin 300130, China; 4State Key Laboratory of Reliability and Intelligence of Electrical Equipment, Hebei University of Technology, Tianjin 300130, China; 5Mechanical and Materials Engineering, University of Nebraska-Lincoln, Lincoln, NE 68588, USA; 6School of Materials Engineering, Purdue University, West Lafayette, Indiana, IN 47907, USA; 7School of Material Science and Engineering, Lanzhou University of Technology, Lanzhou 730050, China

**Keywords:** He ion irradiation, cavities, interface, nanostructured materials

## Abstract

In advanced fission and fusion reactors, structural materials suffer from high dose irradiation by energetic particles and are subject to severe microstructure damage. He atoms, as a byproduct of the (n, α) transmutation reaction, could accumulate to form deleterious cavities, which accelerate radiation-induced embrittlement, swelling and surface deterioration, ultimately degrade the service lifetime of reactor materials. Extensive studies have been performed to explore the strategies that can mitigate He ion irradiation damage. Recently, nanostructured materials have received broad attention because they contain abundant interfaces that are efficient sinks for radiation-induced defects. In this review, we summarize and analyze the current understandings on interface effects on He ion irradiation in nanostructured materials. Some key challenges and research directions are highlighted for studying the interface effects on radiation damage in nanostructured materials.

## 1. Introduction

### 1.1. Motivation and Architecture

As a reliable, sustainable and affordable energy, nuclear power provides more than 13% of electricity worldwide [1]. The next generation nuclear reactors have increasing demands for the discovery of advanced materials that can survive severe radiation environments, so that nuclear reactors can operate safely for a prolonged period of time. In nuclear fission and fusion reactors, helium (He) ions are produced by the decay of tritium and the (n, α) reaction during neutron irradiation [2,3]. He is insoluble in most nuclear materials and its diffusion activation energy is low, so it is easy to form interstitials and He bubbles [4]. He can combine with cavities and accelerate irradiation-induced swelling, hardening, embrittlement and surface deterioration [5,6]. Prior studies show that nanostructured materials possess outstanding irradiation resistance because abundant interfaces can hinder the formation of cavities [7]. However, the underlying mechanisms for interfaces enhanced irradiation resistance had not been revealed in-depth in the past. Recently, many achievements have been made to explore the interface mechanisms for designing suitable nuclear materials. To understand the interface mechanisms and provide a theoretical basis for designing nuclear materials, we summarize these research achievements in this review.

In this review, we first introduce He ion irradiation damage in materials. Then, we summarize the irradiation response in nanostructured materials, including the interactions between cavities and interfaces, and interface effects on irradiation damage. Finally, we discuss the evolutions of properties of irradiated nanostructured materials.

### 1.2. He Ion Irradiation Behavior

#### 1.2.1. He Diffusion 

The solubility of He in most nuclear materials is negligible [8], and the accumulation of He atoms in materials after irradiation is usually accompanied by abundant defect clusters. In addition, the way of He diffusion depends on the dose of irradiation, temperature as well as the presence of other intrinsic or irradiation-induced defects, such as point defects (interstitial and vacancy), interstitial clusters, He-vacancy clusters, cavities, acting as traps for He atoms. Regardless of the way of He diffusion, thermal activation is the most important basic process [8].

Figure 1 shows different means for He diffusion at low radiation damage level [8]:

(1) Interstitial migration: He atoms at interstitial sites diffuse interstitially. Since the activation energy of the He atoms interstitial migration is only a few tenths of eV [8], the He atoms can diffuse quickly at room temperature until they are trapped by other defects. In the case of a low radiation dose, the concentration of defects introduced by irradiation is negligible low, hence He atoms diffuse effectively as interstitials at T < 0.5T_m_ (T is irradiation temperature, T_m_ is melting point of materials), where both the displacement damage and the concentration of thermal vacancies are small. Therefore, interstitial migration is the most probable type of He diffusion mechanisms.

(2) Vacancy mechanism: He atoms exchange position with the neighbor vacancies. When T > 0.5T_m_, abundant vacancies would form by thermal excitation, hence He atoms could jump from one to the other vacancy [9].

For high dose irradiation, He atoms would accumulate to form defect clusters, such as interstitial clusters, He-vacancy clusters, cavities (bubbles and voids), which display different diffusion mechanisms: when T < 0.2T_m_, the athermal “displacement or cascade mechanism” would be the dominant mechanism for He diffusion, because vacancies are almost immobile; when 0.2T_m_ < T < 0.5T_m_, the “replacement mechanism” would be the dominant mechanism—at this time a He atom diffuses interstitially from a vacancy, then it can be re-trapped by another vacancy; when T > 0.5T_m_, the “vacancy mechanism” would be dominant, because He diffusion can be assisted by numerous vacancies (formed by thermal activation) that can move easily.

#### 1.2.2. Nucleation of He Bubbles

Before introducing the bubble nucleation, it is necessary to distinguish the forms of cavities, i.e., bubble and void. Usually, bubble is a kind of spherical cavity, possessing a high concentration of He atoms, hence the bubble has over-pressure or equilibrium-pressure. However, a void with plenty of vacancies has internal pressure much lower than equilibrium-pressure, so that the void has a faceted structure composed of close-packed planes [10]. For cavity formation, first He atoms and vacancies agglomerate into clusters, which accelerate the bubble nucleation and growth. Then with the increasing of irradiated dose or temperature, bubbles would achieve a critical radius via absorbing more He atoms and vacancies and transform into voids [11]. Consequently, bubble and void are two distinct forms of cavities, but with a strong connection. 

Here, we focus on bubble nucleation. There are two types of nucleation of bubbles: homogeneous and heterogeneous nucleation [8]. Homogeneous nucleation means that bubbles nucleate in perfect lattice position, while heterogeneous nucleation preferentially occurs on defects of materials, such as dislocations and interfaces. These two ways are competing during actual nucleation process in materials suffering He ion irradiation.

At low temperatures, thermal dissociation of He atoms from their traps is negligible. Homogeneous nuclei and preexisting defects in materials are competitive for absorbing He atoms. If nuclei can capture more He atoms, in other words, the density of nuclei is higher than that of preexisting defects in the material, homogenous nucleation will dominate (usually at low temperatures or high He implantation doses).

At high temperatures, the thermal dissociation of He atoms from captured positions should be taken into account, and the relation between capture abilities of possible nucleation sites could not explain clearly which nucleation mode dominates. Some defects, such as dislocations and interfaces, have impacts on the thermodynamics of critical nuclei. Figure 2 shows the traditional understanding of how interfaces influence the thermodynamic state of a nucleus [8]. For instance, bubbles are in thermal equilibrium, and the relation among the gas pressure inside bubbles (p), the surface energy of bubbles (γ) and the radius of bubbles (r) can be written as [8]:
(1)p=2γ/r

With the same nucleus volume, the interfacial equilibrium between surface of a bubble and grain boundary (GB) can be seen in Figure 2b, and the equilibrium between a bubble and a GB-precipitate can be seen in Figure 2c.

Radius of the bubble at a GB is bigger than that in the grain interior, thus reducing gas pressure and equilibrium concentration of He atoms. In other words, the critical He concentration for heterogeneous nucleation at a GB is smaller than that for homogeneous nucleation. Therefore, a bubble finds it easy to nucleate around a GB where the nucleation barrier is low.

#### 1.2.3. The Coarsening of Bubbles

Apparently, bubbles tend to grow into voids, namely bubble to void transformation. After nucleation, bubbles can continue to grow by capturing He atoms and vacancies. In addition, the coarsening of the bubble is influenced by several parameters, such as irradiation temperature, He concentration and irradiation time. As the temperature increases, more vacancies are activated and the probability of combining He atoms and vacancies increases, so that the rate of bubble to void transformation increases. In a similar way, higher He concentration and longer irradiation time, produce bigger voids in materials. There are two classic coarsening mechanisms as shown in Figure 3: (1) migration and coalescence (MC) [12]; (2) Ostwald ripening (OR) [13].

(1) Migration and coalescence: bubble migration is because of random rearrangements of the bubble surface via diffusion of matrix atoms. These processes can be achieved by three ways: bulk diffusion, surface diffusion and vapor transport. For the bulk diffusion process, atoms can migrate from the front of a bubble into the matrix, reaching the back of the bubble through body diffusion. Surface diffusion is due to random rearrangements of the bubble surface by diffusion of matrix atoms [8]. Vapor transport means that matrix atoms move into a bubble in the form of vapor, thereby changing its shape and causing it to migrate. Usually, the migration of bubbles is random, that is to say, Brownian motion. Small bubbles will migrate and coalesce into larger bubbles and voids by various ways. Different from bubbles, void migration depends on surface ledge nucleation significantly [14]. This mechanism has been demonstrated in vanadium by in situ hot stage transmission electron microscopy (TEM), where the voids migrate in a random walk manner at 0.55T_m_ [15].

(2) Ostwald ripening: large bubbles can coarsen by capturing and absorbing He atoms, which are dissolved by thermal activation from other small bubbles. 

Temperature is a key parameter that determines coarsening mechanisms. The OR mechanism dominates at high temperatures, suggesting that the bubble dissociation needs higher thermal activation energy than that for driving bubble migration. The energy driving bubble dissociation is not high enough at low temperatures, therefore, the growth of bubbles at low temperatures depend strongly upon the MC mechanism, especially upon the surface diffusion [16]. Whatever mechanisms He bubbles grow by, the number of bubbles continues to decrease with the growth of bubbles. In addition, the mechanism about the growth of He bubbles is very complicated with presence of dislocations, interfaces and other defects. Under high temperature creeping, dislocations motion and GBs sliding can sweep a He bubble, thus promoting the growth of this bubble [17].

### 1.3. The Influence of He Ion Irradiation on Mechanical Properties

As inert gas atoms, He atoms are nearly insoluble in nuclear materials. They usually diffuse and aggregate interstitially and combine with vacancies or cavities. Cavities can continue to migrate and grow. This process can lead to hardening, swelling, embrittlement and surface deterioration, hence degrading mechanical properties.

Since He ion irradiation can induce abundant bubbles which hinder dislocation motion severely, hardening often takes place in irradiated materials. Yang et al. conducted the same irradiation experiments at room temperature for Ni, 304 stainless steel and CrMnFeCoNi, which have similar hardening behavior despite the different compositions these materials possess [18]. Similar results also appeared in nanolayered composites, for example, most multilayers suffered hardening after He ion irradiation at room temperature [19,20,21,22].

Bubbles can grow continuously by absorbing He atoms and vacancies to form large voids, thus making materials swell. Wakai et al. studied swelling behaviors in Ni alloys (Ni–Si alloys, Ni–Co alloys and so on) by high dose irradiation experiments [23]. They found the percent of swelling increases with the size and number density of voids increasing in these Ni alloys. In addition, when pure Ni was irradiated with a dose of $5\;\times \;10^{17}\;\mathrm{ions}/{\mathrm{cm}}^{2}$ and annealed at 973 K, the swelling was more than 6% [24].

With numerous big voids forming in materials, embrittlement and surface deterioration take place inevitably. Shinohara et al. did a series of tensile experiments at different temperatures for unirradiated and irradiated pure Fe (where samples were irradiated using 100 keV He ions with a dose of $1\times 10^{17}{\;\mathrm{ions}/\mathrm{cm}}^{2}$) [25]. They found the ductile–brittle transition temperature for irradiated polycrystalline Fe is higher than that for unirradiated Fe. Although irradiation-induced defects are within 1 µm of the surface, bulk samples show more embrittlement after irradiation. About surface deterioration, Yoshida, et al. found the surface of W samples transformed from straight to wavy after irradiation [26].

### 1.4. Strategy for Irradiation Resistance: Interfaces

Generally, He ion irradiation damage results from numerous point defects (interstitials and vacancies) induced by the irradiation process. The point defects can aggregate and form cavities (bubbles and voids) which degrade materials severely [4]. Interfaces, as typical planar defects, whether homogenous GBs or heterogeneous phase interfaces, can attract, absorb and annihilate point defects efficiently [5,6].

For homogenous GBs, Bai et al. found GBs have a surprising “loading–unloading” effect during He ion irradiation [27]. During irradiation, interstitials were absorbed into GBs via the loading process, then GBs as a source emitted interstitials to annihilate vacancies in the bulk near GBs. This GB assisted process has a much lower energy barrier compared with conventional defects diffusion. Also, this mechanism is efficient for annihilating irradiation-induced defects in the bulk, resulting in self-healing of the irradiation damage.

For heterogeneous interfaces, taking Cu/Nb interfaces for example, there are numerous misfit dislocations at Cu/Nb interfaces because of lattice mismatch, and these dislocations can form misfit dislocation interactions (MDIs). Also, these MDIs can absorb and trap point defects efficiently and induce ‘‘platelet-to-bubble’’ transition (Figure 4) at interfaces during irradiation [28]. This transition is governed by a competition between three kinds of pressures acting on interfacial He-filled bubbles: the mechanical pressure p_He_ of the trapped He gas, the osmotic pressure p_V_ due to the flux of radiation-induced vacancies within the crystal to the bubble, and the capillary pressure pc arising from the surface energy of the bubble. p_He_ and p_V_ tend to expand the bubble while pc tends to shrink it. If these three pressures balance, like this [28]:(2)$$p_{He}+p_{V}=p_{C}$$
then, the bubble is in equilibrium: it neither expands nor contracts. 

At the same irradiation condition, a spherical bubble, ~2 nm in diameter, forms within a crystalline solid, while a platelet-shaped He-filled bubble generates at an interface. Also, the volume of such platelets was nearly three times smaller than that of bubbles in face-centered cubic (FCC) Cu with the same number of He atoms [29]. Thus, platelets store He atoms more efficiently than spherical bubbles and leads to less He-induced hardening and swelling prior to void formation. Therefore, as platelets do not degrade materials as much as bubbles, heterogeneous interfaces can make materials possess more irradiation resistance.

## 2. Irradiation Response in Nanostructured Materials

### 2.1. The Interactions between Cavities and Interfaces

Although interfaces can absorb, trap and annihilate defects efficiently during He ion irradiation, cavities (bubbles and voids) form in most nanostructured materials as the implantation dose is enough. However, interfaces can still tune cavities. There are two types of interactions between interfaces and cavities. First, cavities can stay away from interfaces and generate void-denuded zones [30,31,32]. Second, cavities can adhere to interfaces or cross interfaces [22,32,33]. There are many factors that tailor the formation of cavities, such as implantation dose, temperature and layer constitution. Also, a bubble will transform into a void as it grows, thus bubble and void are at different stages during the growth of a cavity. Once a cavity forms, and the interaction between it and the interface does not change with the growth of this cavity. Therefore, we will take the void as an example and discuss void-interface interactions.

#### 2.1.1. Void-Denuded Zones near Interfaces

A void-denuded zone (VDZ) can generate in accumulative roll bonding (ARB) Cu/Nb nanolayered composites when the irradiation experiment is designed like Figure 5a [31]. When He ions are implanted parallel to the interfaces, the interfaces can absorb defects induced by irradiation efficiently and make defect concentration lower near the interfaces than that within layers. The difference in defect concentration between layer interiors and interfaces leads to long-range diffusion of defects toward interfaces. Once a steady state is established, the defect concentration monotonically increases from interfaces to the interiors of layers, thus inducing the density gradient of voids which is controlled by the diffusion process [30]. In other words, the density of voids in the layers is higher than that near the interfaces. It can be proved by counting the number of voids in different areas within the Cu layer (Figure 5b,c). However, different interfaces may have different abilities for sinking defects induced by irradiation, therefore, they may form different gradients for defect concentration and generate VDZs with different widths. These abilities can be estimated by the widths of VDZs (Figure 5d). In other words, the wider VDZs are, the better sinking efficiency interfaces have. For example, Figure 5e shows widths of VDZs for different Cu/Nb interfaces in 135 nm ARB Cu/Nb, such as Cu {112}//Nb {112} interfaces, Cu {111}//Nb {110} interfaces and so on. The widths of VDZs are measured by TEM pictures and the orientation relationships of the interfaces are confirmed by precession electron diffraction (PED). According to these widths, we can find these atomic-ordered Cu {112}//Nb {112} interfaces have the best sinking abilities for defects.

#### 2.1.2. Voids Adhere to Interfaces or Cross Interfaces

Voids can adhere to interfaces or cross interfaces. In Figure 6a,b, voids adhere to interfaces in bulk Cu/Ag and ARB Cu/Nb nanolaminates irradiated using 200 keV He ions with a dose of $2\times 10^{17}{\;\mathrm{ions}/\mathrm{cm}}^{2}$ at 450 °C. However, voids adhere to interfaces on the side of the Ag layer in bulk Cu/Ag and in the Cu layer for ARB Cu/Nb [22,33]. This asymmetrical void–interface interaction is a consequence of different surface energies of the two metals and non-uniformity in interface energy [33]. To confirm this explanation further, first, we introduce void–interface interaction on twin boundaries (TBs) as shown in Figure 6c. TBs possess uniform interface energy and the matrix and twin have the same surface energy, therefore voids cross TBs symmetrically. Second, we take bulk Cu/Ag as an example and use an interface wetting model (Figure 7a) to explain this asymmetrical interaction. The equation regarding a wetting model can be written as [33]:(3)$$W=\gamma_{A-C}+\;\gamma_{A-B}-\;\gamma_{B-C}$$
where $\gamma_{A-C}$, $\gamma_{A-B}$ and $\gamma_{B-C}$ are surface energy of A–C, A–B and B–C interface. In Figure 7b, if W > 0, voids will stay in the A phase and touch the interface because wetting is favored. In contrast, if W < 0, wetting is not favored, voids should have minimum energy and stay within the phase (the A phase) with the lower free surface energy entirely. In addition, there are abundant MDIs at Cu/Ag interfaces (Figure 7c). Molecular dynamics (MD) simulation shows that W > 0 at MDIs and W < 0 at coherent parts of the interface between MDIs (Figure 7d), therefore, voids only wet MDIs during irradiation. Once a void has grown large enough to cover an entire MDI, it is not thermodynamically favorable for this void to continue to wet the interface regions where W < 0. Instead, the void prefers extending into the Ag layer with lower surface energy.

Irradiation experiment parameters, such as temperature and layer thickness, can also determine whether voids adhere to interfaces or cross interfaces [22]. For 58 nm ARB Cu/Nb, voids do not cross the interfaces independent of implantation dose at room temperature, but voids can overlap interfaces at 450 °C (Figure 8a,c,e). The increase of irradiation dose can only induce the growth of voids, and high irradiation temperature can not only increase the size of voids but also make voids cross interfaces. For 16 nm ARB Cu/Nb, voids always adhere to the interfaces from the Cu layer and never cross interfaces despite irradiation temperature (Figure 8b,d). Layer thickness is another key point for the distribution of voids. ARB Cu/Nb interfaces have excellent abilities for irradiation resistance, therefore voids are confined within Cu layers despite the temperature for 16 nm ARB Cu/Nb with a high interface density.

### 2.2. Interfaces Effects on Irradiation Damage

Cavity is a basic form of irradiation production, and its diameter, density and distribution can significantly influence irradiation damage during He ion irradiation. Also, effects of cavity on irradiation damage can be tailored by interfaces. Interfaces can be divided into incoherent interfaces, semi-coherent interfaces and coherent interfaces [34,35]. When lattice mismatch on a boundary is large, and arrays of misfit dislocations and elastic deformation on the boundary cannot accommodate this strain, the boundary is an incoherent interface. However, when arrays of misfit dislocations can mediate this strain, the boundary is a semi-coherent interface. When elastic deformation can accommodate this strain, the boundary is a coherent interface. Next, we will discuss the interface effects on cavity induced irradiation damage.

#### 2.2.1. Incoherent Interfaces

Most GBs are typical incoherent interfaces, and they are believed to have excellent abilities for sinking defects due to the surprising “loading–unloading” effect which has been illustrated in the previous chapter. Also, this ability for sinking defects (sink strength) is related to grain size for the same materials. To describe the relationship between sink strength and grain size, Bullough et al. [36] used the cellular model and acquired this relationship:(4)$$k_{gb}^{2}=15/R^{2},$$
where $k_{gb}^{2}$ is the sink strength for grain boundary and *R* is grain size. From this equation, we can find that sink strength increases with grain size decreasing.

Nanocrystalline (NC) materials have good irradiation resistance compared to their coarse-grained (CG) counterparts because of large sink strength for irradiation induced defects. For example, austenitic Fe–Cr–Ni ternary alloys fabricated by vacuum cast and subsequent hot isostatic pressing have coarse grains with a grain size of 700 μm (Figure 9a). The alloys display ultrafine grains (UFG) with a grain size of 400 nm after equal channel angular pressing (ECAP) up to eight passes at 500 °C (Figure 9b). In Figure 9c,d, when CG and UFG Fe–Cr–Ni alloys are irradiated together using 100 keV He ions to a dose of $6\times 10^{16}{\;\mathrm{ions}/\mathrm{cm}}^{2}$, UFG Fe–Cr–Ni alloys have less bubbles compared to CG Fe–Cr–Ni alloys (Figure 9e). Also, the average bubbles size in UFG Fe–Cr–Ni alloys is 1.0 nm, which is smaller than that in CG Fe–Cr–Ni alloys (Figure 9f) [37].

In situ irradiation experiments also have been conducted in order to confirm this superior irradiation tolerance of GBs. NC Fe films fabricated by sputter deposition experienced in situ He ion irradiation with a total dose of $2.8\times 10^{21}{\;\mathrm{ions}/m}^{2}$ in TEM. Figure 10 shows TEM pictures of the samples after irradiation and the profile plotted with average void density vs. grain size (area). During this irradiation experiment, the appearance of void-denuded zones was recorded accurately in videos. Also, the smaller grain diameter resulted in less void density [38]. These results prove that GBs have superior sinking abilities and NC materials can alleviate irradiation damage effectively.

#### 2.2.2. Semi-Coherent Interfaces 

Most multilayers fabricated by physical vapor deposition (PVD) have semi-coherent interfaces, such as PVD Cu/Nb, Cu/V and Cu/Mo. Their interface structures are similar, for instance, PVD Cu/Nb, Cu/V and Cu/Mo have atomic straight FCC {111}//BCC {110} interfaces (where FCC is face-centered cubic and BCC is body-centered cubic). These multilayers also have superior irradiation resistance due to abundant MDIs at interfaces which can trap defects effectively and induce “platelet-to-bubble’’ transition during irradiation. Also, the sink strength is related to layer thickness for the same multilayer, and the relationship between sink strength and layer thickness can be described as [39]:(5)$$k_{h}^{2}=12\eta /h^{2},$$
where $k_{h}^{2}$ is the layer interface sink strength, *h* is layer thickness and *η* is a constant for the multilayer. When lattice mismatch on the layer interface is large, *η* is also large.

There are many experimental results [7,19,40,41,42,43] which are consistent with what these theories predict. When PVD Cu/V and Cu/Mo were irradiated using He ions with a dose of $6\times 10^{16}{\;\mathrm{ions}/\mathrm{cm}}^{2}$, the diameters of bubbles were all about 1 nm (Figure 11a,b), and the densities of bubbles all decreased with layer thickness reducing because the sink strengths increased [19,40]. Moreover, PVD Cu/Nb has the largest sink strength in these multilayers with the same layer thickness, because lattice mismatch on the Cu/Nb interface is highest according to O-lattice theory [44,45]. For PVD Cu/Nb with 100 nm layer thickness, the size of bubble was 1 nm (Figure 12a,b) when it was irradiated by He ions with a dose of $1\times 10^{17}{\;\mathrm{ions}/\mathrm{cm}}^{2}$ [7], although this implantation dose is higher than the PVD Cu/V and Cu/Mo experienced. Also, when 2.5 nm PVD Cu/Nb experienced the same irradiation, bubbles were not observed independently of defocused conditions of TEM (Figure 12c) [7]. In other words, high density of semi-coherent interfaces, which have the superior sink strength for irradiation-induced defects, can hinder bubble formation significantly in 2.5 nm PVD Cu/Nb.

Semi-coherent interfaces also have superior irradiation resistance at high temperatures. For example, when 120 nm PVD Cu/Nb experienced He ion irradiation with a dose of $1\times 10^{17}{\;\mathrm{ions}/\mathrm{cm}}^{2}$ at 490 °C, voids generated in the Cu layer and displayed a size range between 4.5 and 43 nm (Figure 13a). In contrast to the Cu layer, bubbles formed in the Nb layer and their diameter was 1 nm [46]. For 5 nm PVD Cu/Nb, voids in Cu layers spanned the whole Cu thickness and elongated between 7 and 11 nm, while the size of bubbles in Nb layers was still about 1 nm (Figure 13b). From these results, we can see bubbles coarsen rapidly and voids form in the Cu layer, and bubbles in the Nb layer do not change at the elevated temperatures. The reasons for this phenomenon are as follows: first, T > 0.5T_m_ in the Cu layer, the migration of vacancies is easy, and bubbles coarsen rapidly and transform into voids by absorbing interstitials and vacancies. Second, T < 0.5 T_m_ in the Nb layer, the migration of vacancies is not as easy as that in Cu and bubbles coarsen slowly by absorbing interstitials. Although bubbles grow rapidly and voids form inevitably in the Cu layer at high temperatures, semi-coherent interfaces can confine big voids within the layers and control the distribution of cavities. Also, the size of voids decreases with the increasing density of interfaces.

Besides layer interfaces in multilayers, some ferrite/precipitate interfaces (ferrite/Y_2_Ti_2_O_7_ interfaces) are also semi-coherent interfaces in the nano-sized oxide dispersion strengthened (ODS) steels [47]. These interfaces can trap interstitials and vacancies efficiently and prevent abundant vacancies and He from entering ferrite. Therefore, ODS steels, which have abundant ferrite/oxide interfaces, possess excellent irradiation resistance. To explain this ability of trapping defects, Yang et al. [48] conducted structural relaxations and energetic calculations using the density functional theory (DFT) for ferrite/Y_2_Ti_2_O_7_ interfaces in ODS steels. They found these interfaces have the cube-on-cube orientation relationship of {100} < 100>_Fe_// {100} < 100> _Y2Ti2O7_ and enrich Yi and Ti. Also, the interfacial He and vacancy formation energies are lower at the ferrite/Y_2_Ti_2_O_7_ interfaces than at ferrite matrix and GBs as predicted in Reference [48].

In order to prove superior irradiation resistance of ODS steels, some irradiation experiments were also performed. For example, when 14Cr-ODS and Eurofer 97 steel experienced He ion irradiation with a dose of $1\times 10^{17}{\;\mathrm{ions}/\mathrm{cm}}^{2}$ at 400 °C [49], the average bubble size at the peak depth was 3.6 nm in 14Cr-ODS, and 5.2 nm in Eurofer 97 (Figure 14a–c). Also, the maximum bubble volume fraction in 14Cr-ODS was 0.5%, and 1.1% in Eurofer 97 (Figure 14a,b,d). Compared to Eurofer 97, 14Cr-ODS contains abundant nano-sized Y_2_Ti_2_O_7_ precipitates which are identified in Figure 11e,f. Therefore, these dominant ferrite/Y_2_Ti_2_O_7_ interfaces lead to smaller bubble size and less bubble volume fraction.

#### 2.2.3. Coherent Interfaces 

Coherent twin boundaries (CTBs) are typical fully coherent interfaces. During He ion irradiation, CTBs are believed to not possess as strong an ability for sinking irradiated defects as normal GBs, the reasons are as follows [50]: first, the excess free volume is low near CTBs; second, atomistic simulations find the vacancy and interstitial formation energies at CTBs are nearly identical to that at the interior of grains. 

Nanotwined (NT) materials despite possessing high density of CTBs, however, cannot resist He ion irradiation damage efficiently. Taking NT Cu films as an example, we illustrate this weak ability of irradiation resistance for NT materials further [50]. In Figure 15, when experiencing He ion irradiation at a dose of $1\times 10^{17}{\;\mathrm{ions}/\mathrm{cm}}^{2}$, the diameter of bubbles in NT Cu films is about 2 nm, which is larger than that in PVD Cu/Nb whose layer thickness is similar to the twin spacing. Bubbles can be observed throughout all implanted regions, even within 5 nm of the surface where He concentration is very low (<$1\times 10^{16}{\;\mathrm{ions}/\mathrm{cm}}^{2}$). Also, the density of bubbles in NT Cu films is similar to that in coarse-grained Cu under the same irradiation parameter.

To confirm this limited effect of CTBs on He ion irradiation resistance, some studies have compared the irradiation response of GBs and CTBs for He ion irradiation directly [51]. In their work, TEM samples of Cu which have GBs and CTBs were irradiated with a dose of $1\times 10^{17}{\;\mathrm{ions}/\mathrm{cm}}^{2}$. Then, the distribution of cavities around GBs and CTBs was acquired. In Figure 16a, the density of cavities is lower near GBs and cavity-depleted zones (like void-denuded zones) along the GBs are readily observed. However, cavities are almost homogenously distributed across the CTBs (Figure 16b), suggesting that CTBs do not trap defects efficiently and hinder cavity formation compared to GBs. Also, some zones can be selected to evaluate the interface effects for irradiation resistance quantitatively. In Figure 16c, zone 1 and zone 2 around interfaces are chosen (GB and CTB) for counting cavities, and the statistical results are shown in Figure 16d. For GBs, the number of cavities in zone 2 is about three times of that in zone 1. However, for CTBs, the number of cavities in zone 1 and zone 2 is similar. It means that GBs can alleviate He ion irradiation resistance efficiently and lower the density of cavities locally (about two thirds) and CTBs do not have the same strong ability compared with GBs.

Some multilayers also have abundant coherent interfaces because lattice mismatch along their interface can be accommodated by coherency strain. Compared to twin boundaries, these heterogeneous coherent interfaces have better abilities for resisting He ion irradiation. The reasons can be summarized as follows: first, coherent interfaces have coherent stress which can promote vacancies and interstitials migration to interfaces [52]; second, coherent interfaces are interrupted by irradiation induced defects and generate disconnections which can absorb defects further; third, once bubbles nucleate on the interfaces, coherency stress can hinder bubble growth.

To prove the good sink ability for heterogeneous coherent interfaces, PVD Cu/Fe with 0.75 nm layer thickness is taken for an example [20]. PVD Cu/Fe with 0.75 nm has numerous Cu {111}//Fe {110} interfaces according to X-ray diffraction (XRD) analysis. After irradiation using 100 keV He ions with a dose of $6\times 10^{16}{\;\mathrm{ions}/\mathrm{cm}}^{2}$ at room temperature, the diameter of bubbles is about 1 nm (Figure 17a), which is comparable to that in PVD Cu/V and Cu/Mo experienced the same He ion irradiation. Also, although these Cu/Fe interfaces are coherent, cavities are still confined by interfaces (Figure 17b) and their diameter is curtailed with layer thickness reducing (Figure 17c). In conclusion, nanostructured materials with heterogeneous coherent interfaces can also resist irradiation damage efficiently.

## 3. The Evolution of Mechanical Properties of Irradiated Nanostructured Materials

After irradiation, the mechanical properties of nanostructured materials change due to the formation of cavities. However, these changes can be tuned by interfaces. Next, taking hardening and softening behaviors as examples, we will discuss these interface-related changes of properties in detail.

### 3.1. Hardening Behavior: Small Cavities

Compared to their large-sized counterparts (where size represents grain size in NC materials, layer thickness in nanolayered composites and so on), smaller cavities can form in nanostructured materials after the same He ion irradiation at room temperature because of interface effects. These small cavities, like nanoscale precipitated phases, can block dislocation motion severely, thus contributing to hardening in nanostructured materials. Also, irradiation dose and the size of the nanostructured materials, which determine the density of small cavities, can influence the hardening rate. These influences have been investigated widely by hardness measurement before and after irradiation [22,43,53,54,55,56]. As shown in Figure 18, we can see hardening is directly proportional to irradiation dose for PVD Cu/V [53]. Also, hardening becomes weak with layer thickness decreasing for PVD Ag/V, Ag/Ni and Cu/V [43,54,55], and it becomes strong with layer thickness decreasing for PVD Cu/Co [56]. In other words, there are two types of size effects on irradiation-induced hardening in nanostructured materials. Next, taking PVD Ag/V and Cu/Co as examples, we explain these size effects.

In PVD Ag/V, the relationship between hardening *∆H* and layer thickness *h* can be described by [43]:(6)$$\Delta H=9\tau_{i}\left(1-\frac{l}{\sqrt{2}h}\right),$$
where τ_i_ is the average shear strength of bubbles, and l is the average obstacle spacing.

Also, τ_i_ which is related to interface can be acquired by the Orowan model and l can be gauged by TEM images. When h > 100 nm, the dislocation pile-up mechanism [57] dominates in as-deposited multilayers. At this length scale, multilayers exhibit some mechanical behavior like bulk. Therefore, h is infinite and ∆H ≈ 9τ_i_. When 5 nm < h < 100 nm, the confined layer slip model [58] dominates in as-deposited multilayers, and cavities appear near interfaces and inside layers which contribute to hardening. At this length scale, ∆H can be directly calculated by the equation above (Figure 19a). When h < 5 nm, the interface crossing mechanism [59] dominates in as-deposited multilayers. At this length scale, a few cavities form because the interfaces of high density can trap and annihilate numerous irradiation-induced point defects (Figure 19b), and these cavities interfere with the crossing dislocations. Under this condition l > h, *∆H* is small and even negligible sometimes.

In PVD Cu/Co, the hardening increases with reducing layer thickness. This abnormal size effect is because interface structures change with decreasing layer thickness. The change of interface structure influences the strengthening mechanism for PVD Cu/Co. In order to explain this size effect in detail, we divide layer thickness into two sections.

When h < 10 nm, Cu/Co interfaces are coherent and the hardness of as-deposited multilayers is dominated by the interface barrier strength to transmission of partial dislocations. After radiation, cavities at the layer interfaces are typically over-pressurized and strong obstacles [19,54,55], thus these partials have to constrict to full dislocations (in Cu layers) before transmitting to the adjacent Co layers (Figure 20). In other words, cavities can strengthen interfaces and block dislocation transmission. At this length scale, cavities-induced hardening is predominant. When 50 nm < h < 200 nm, Cu/Co interfaces are incoherent and the hardness of as-deposited multilayers is determined primarily by high-density stacking faults (SFs) in Co. After irradiation, cavities distribute both along interfaces and within the layers. However, the average cavity spacing l for 100 nm PVD Cu/Co is about 9 nm, which is much larger than the average spacing between SFs (Figure 21). At this length scale, cavity-induced hardening is negligible compared to high density of SFs in multilayers.

### 3.2. Softening Behavior: Large Cavities

When nanostructured materials are irradiated at high temperatures (more than half of the melting point of constituent), cavities can grow rapidly by absorbing abundant interstitials and vacancies. In this case, softening would occur in nanostructured materials after irradiation. For example, hardness of 58 nm ARB Cu/Nb decreased from 3.8 GPa to 3.2 GPa [22] after He ion irradiation at 450 °C with a dose of $2\times 10^{17}{\;\mathrm{ions}/\mathrm{cm}}^{2}$. Also, when NC W experienced He ion irradiation at a dose of $3.6\times 10^{17}{\;\mathrm{ions}/\mathrm{cm}}^{2}$ at 950 °C, its hardness decreased severely from 8.5 GPa to 6.8 GPa [60]. According to TEM images for these irradiated samples (Figure 22), the softening behavior can be explained as follows: first, the size of cavities is large. These cavities are weak obstacles to dislocation and even are sources of dislocations owing to their large surface; second, the large cavities that cross interfaces can destroy interface structures, thus reducing the interface barrier strength to the transmission of dislocations, leading to softening.

## 4. Summary and Outlook

He ion irradiation can induce numerous defects (interstitials and vacancies), and form He bubbles. These bubbles can induce swelling, hardening, embrittlement and surface deterioration, thus degrading mechanical properties substantially.

Nanostructured materials have excellent irradiation responses to He ion irradiation due to high density of interfaces. These interfaces can tune cavities, such as making cavities stay away from interfaces, adhere to interfaces and cross interfaces.

Different interfaces, such as GBs, oxide precipitate interfaces, heterogeneous layer interfaces and TBs, may have different sink strengths for He ion irradiated defects. The He ion irradiated results, like size and density of cavities, can be reduced effectively in nanostructured materials. As for multilayers with semi-coherent interfaces, different parameters, such as lattice mismatch on the interface, layer thickness, irradiation dose and temperature, can influence these results. The specific relationships between these factors and results are concluded in Table 1. Also, the formation of cavities can be suppressed completely. For example, when Cu/Nb multilayers with 2.5 nm layer thickness experienced He ion irradiation with a dose of $1\times 10^{17}{\;\mathrm{ions}/\mathrm{cm}}^{2}$ at room temperature, no cavities formed.

Moreover, after irradiation, both hardening and softening can occur depending on the cavity size. As for multilayers, there are opposite relationships between hardening and layer thickness due to different interface structures.

Although the interface effects on He ion irradiation have been studied extensively recently, there are still numerous subjects that require in-depth exploration.

First, studies on evolution of mechanical properties of irradiated nanostructured materials, such as tensile strength, ductility and so on, are needed. These studies can build relationships between interfaces and application-related properties for nanostructured materials.

Second, more in situ He ion irradiation experiments should be performed for understanding the interactions between interfaces and defects better for different nanostructured materials. Such as in situ TEM study on He ion irradiation can supply direct evidence for defect/cavity formation, migration and interaction with interfaces.

Third, besides interstitials, vacancies and cavities, the interface effects on other defects, like dislocation loops, should also be investigated in-depth. 

## Figures and Tables

**Figure 1 materials-12-02639-f001:**
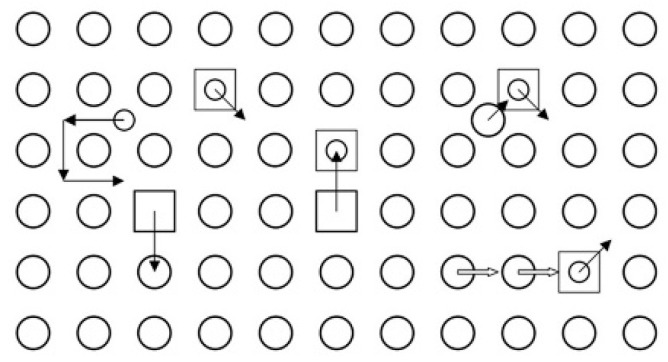
Schematic illustration of defect configurations and jump processes relevant for He diffusion without and with irradiation [8].

**Figure 2 materials-12-02639-f002:**
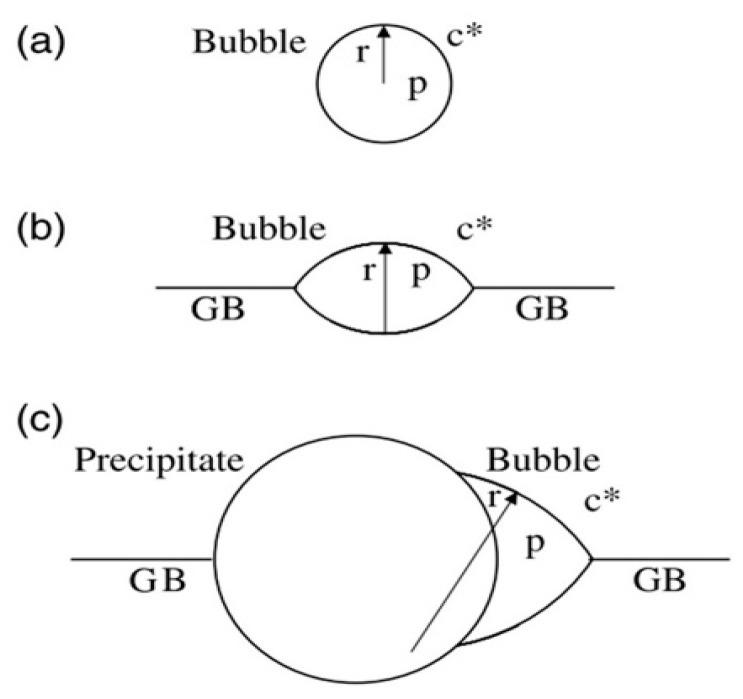
Illustration of interfacial effects on bubble nucleation at high temperatures; (**a**) spherical nucleus in the matrix, (**b**) lenticular nucleus at a grain boundary (GB), (**c**) truncated lenticular nucleus at a GB-precipitate, all assumed to have the same volume [8].

**Figure 3 materials-12-02639-f003:**
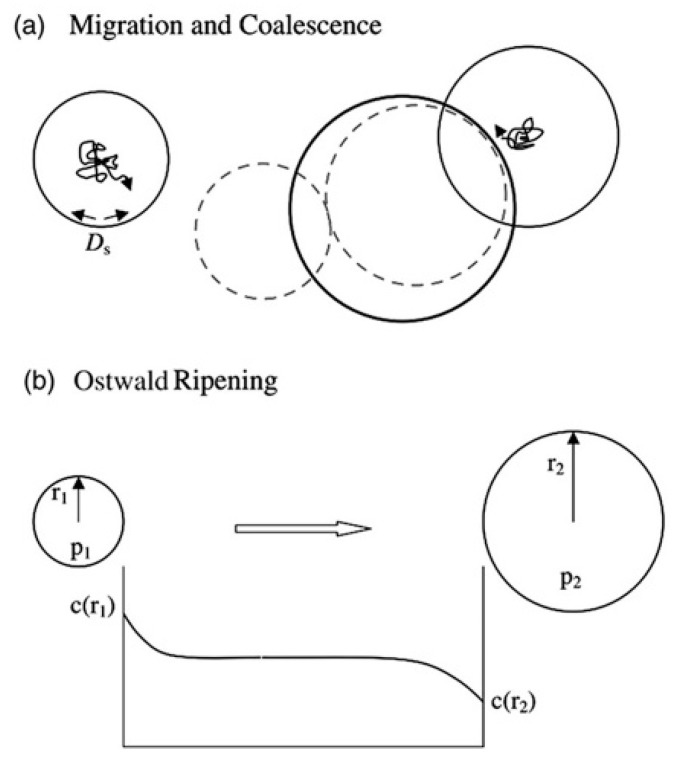
Schematic illustration of the two main bubble coarsening mechanisms, (**a**) migration and coalescence via surface diffusion, (**b**) Ostwald ripening due to He fluxes driven by differences in the thermal equilibrium He concentrations in the vicinity of small and large bubbles [8].

**Figure 4 materials-12-02639-f004:**
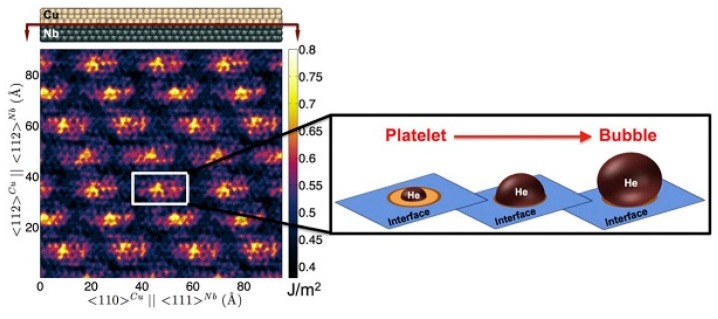
Left: Location-dependent energy of Cu/Nb interfaces. The bright, high-energy regions are heliophilic. Right: A He platelet transforms into a bubble once it has grown to occupy the entire heliophilic patch on which it nucleated [28].

**Figure 5 materials-12-02639-f005:**
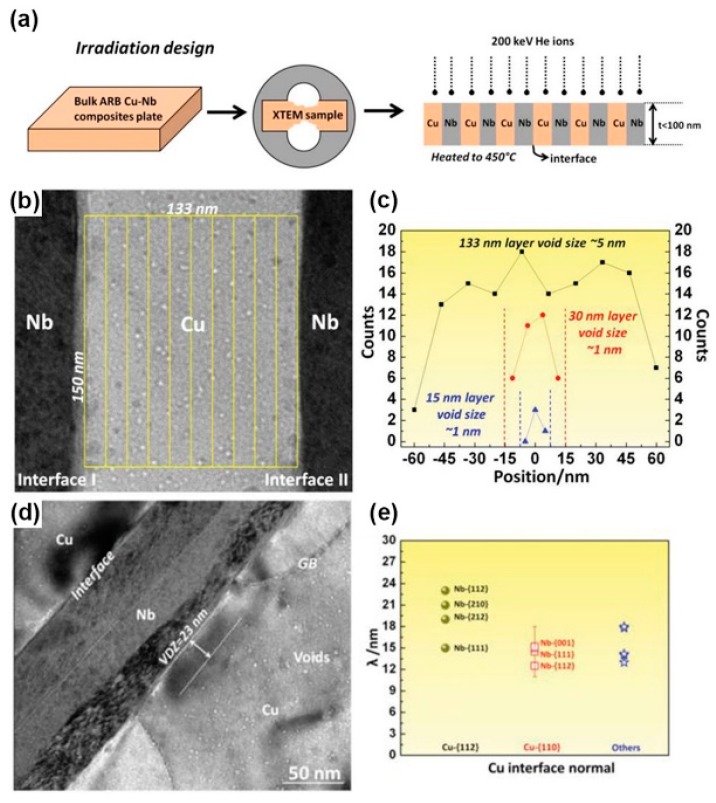
(**a**) Schematic illustration of the He ion irradiation experimental procedures. (**b**) Illustration of the method to determine the void number density in Cu layers. (**c**) Plot of the number density of voids as a function of distance from the center of the layer in 133 nm-, 30 nm- and 15 nm-thick Cu layers. (**d**) Under-focus TEM image showing a void-denuded zone (VDZ) near a Cu/Nb interface. (**e**) Widths of VDZs near Cu/Nb interfaces sorted by their plane orientations [31].

**Figure 6 materials-12-02639-f006:**
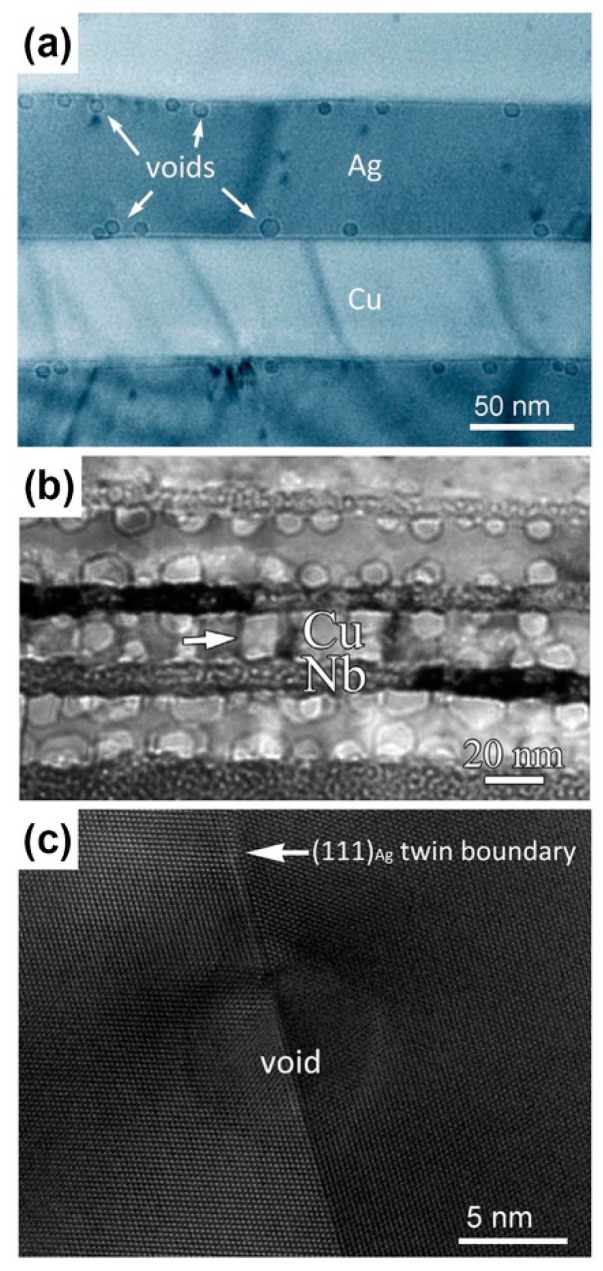
(**a**) Over-focus TEM image of the Cu/Ag composite after He ion irradiation at 450 °C [33]. (**b**) Under-focus TEM image of the ARB Cu/Nb nanolayered composites after He ion irradiation at 450 °C [22]. (**c**) High resolution transmission electron microscopy (HRTEM) image shows void wetting of a coherent (111) Ag twin boundary [33].

**Figure 7 materials-12-02639-f007:**
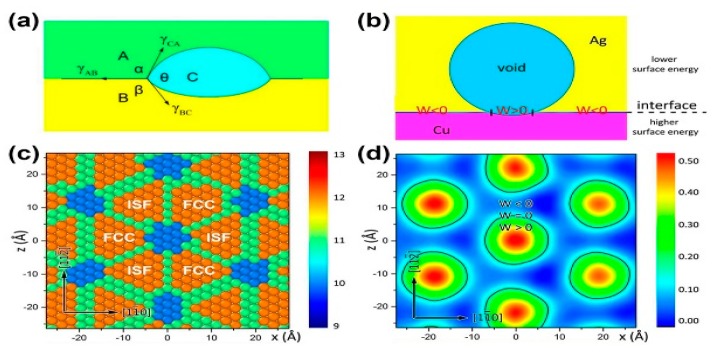
(**a**) Schematic of wetting model on interfaces. (**b**) Schematic of a void wetting a single misfit dislocation interaction (MDI) where W > 0 at a Cu/Ag interface. (**c**) Misfit dislocation network in the cube-on-cube Cu/Ag interface. Atoms shown are on the Cu side of the interface and colored by coordination number. (**d**) Contour plot of the location-dependent interface energy of a Cu/Ag interface. Black contours correspond to zero wetting energy [33].

**Figure 8 materials-12-02639-f008:**
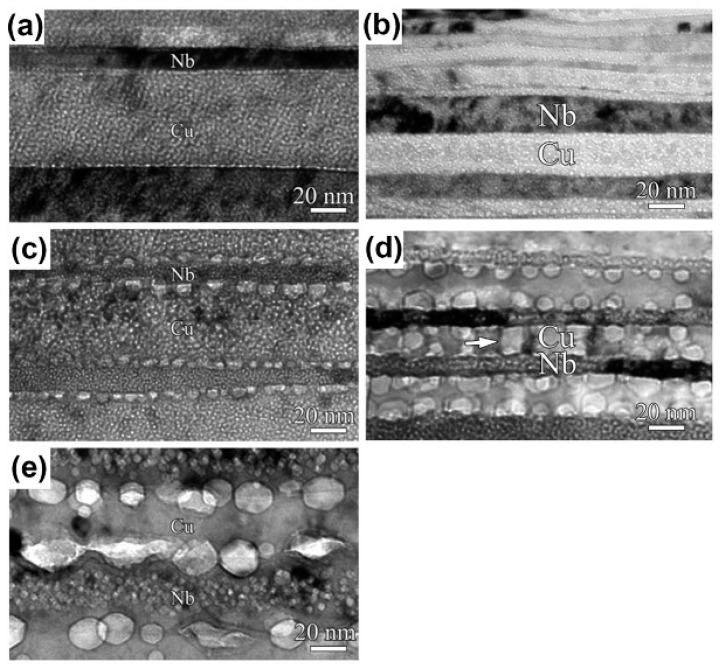
TEM images of accumulative roll bonding (ARB) Cu/Nb after He ion irradiation with different layer thickness, implantation dose and temperature. (**a**) 58 nm, $2\times 10^{17}{\;\mathrm{ions}/\mathrm{cm}}^{2}$, room temperature (RT); (**b**) 16 nm, $2\times 10^{17}{\;\mathrm{ions}/\mathrm{cm}}^{2}$, RT; (**c**) 58 nm, $6.5\times 10^{17}{\;\mathrm{ions}/\mathrm{cm}}^{2}$, RT; (**d**) 16 nm, $2\times 10^{17}{\;\mathrm{ions}/\mathrm{cm}}^{2}$, 450 °C; (**e**) 58 nm, $2\times 10^{17}{\;\mathrm{ions}/\mathrm{cm}}^{2}$, 450 °C [22].

**Figure 9 materials-12-02639-f009:**
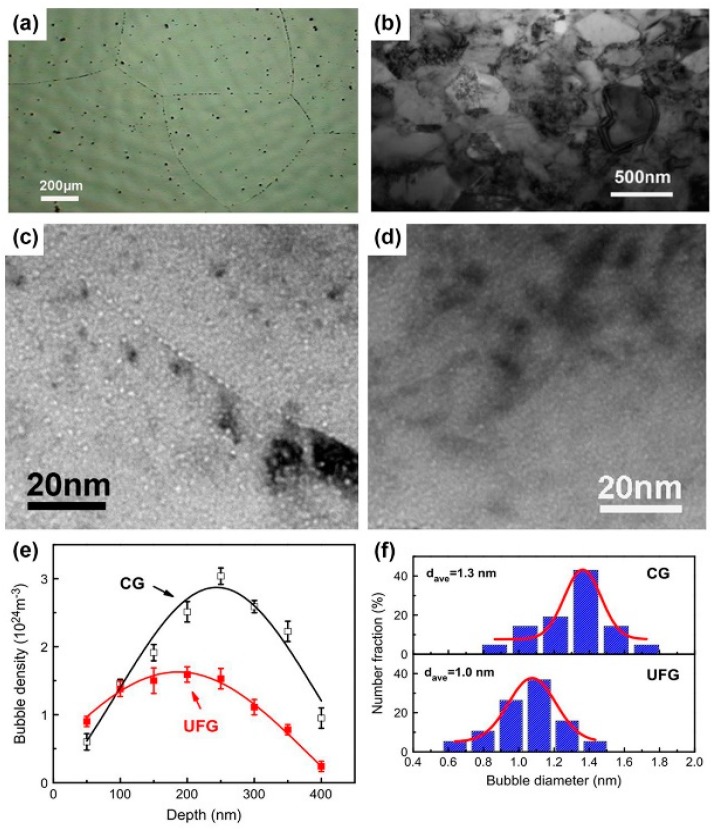
(**a**) An optical microscopy image of as-received coarse-grained (CG) Fe–Cr–Ni alloy. (**b**) TEM micrograph of ultrafine grain (UFG) Fe–Cr–Ni alloy after equal channel angular pressing (ECAP). (**c**,**d**) Under focused TEM micrograph of He ion irradiated CG and UFG Fe–Cr–Ni alloys. (**e**) Depth dependent bubble density in irradiated CG and UFG Fe–Cr–Ni alloy. (**f**) Bubble diameter distribution in the peak damage region of He ion irradiated CG and UFG Fe–Cr–Ni alloys [37].

**Figure 10 materials-12-02639-f010:**
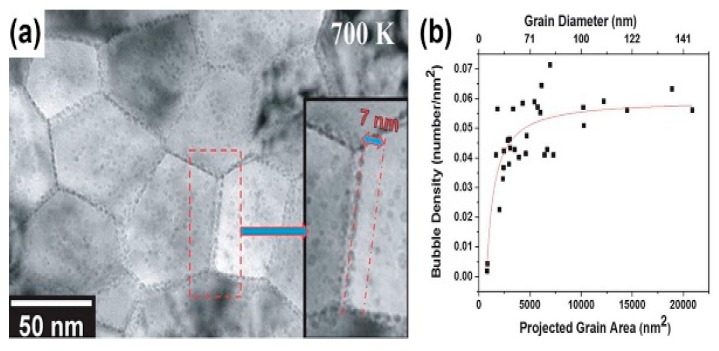
(**a**) Over-focused bright field TEM images of 10 keV He ion irradiation on nanocrystalline iron at calibrated temperatures of 700 K. (**b**) Areal bubble density (number/nm^2^) vs. grain size (area) for 10 keV He ion irradiation on nanocrystalline iron at 700 K [38].

**Figure 11 materials-12-02639-f011:**
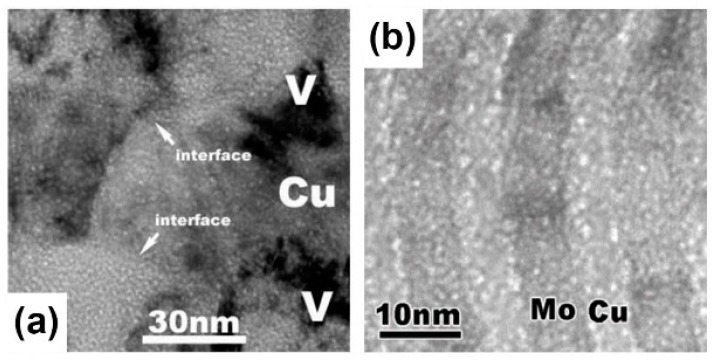
Under focused TEM images of He ion irradiation on multilayers with a dose of $6\times 10^{16}\;\mathrm{ions}/{\mathrm{cm}}^{2}$. (**a**) Cu/V multilayers [19]; (**b**) Cu/Mo multilayers [40].

**Figure 12 materials-12-02639-f012:**
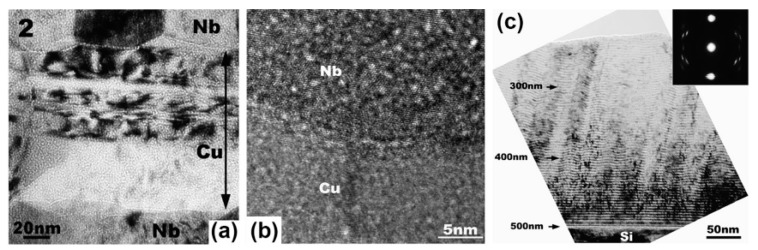
(**a**) Low magnification TEM micrograph of 100 nm Cu/Nb multilayer irradiated with He ions at 150 keV with a dose of $1\times 10^{17}\;\mathrm{ions}/{\mathrm{cm}}^{2}$. (**b**) Defocused HRTEM micrograph of Cu/Nb interfaces indicated cavities in each constituent has 1 nm in diameter and aligned along the interface. (**c**) 2.5 nm Cu/Nb multilayers subjected to He ion irradiation at 150 keV with a dose of $1\times 10^{17}\;\mathrm{ions}/{\mathrm{cm}}^{2}$ [7].

**Figure 13 materials-12-02639-f013:**
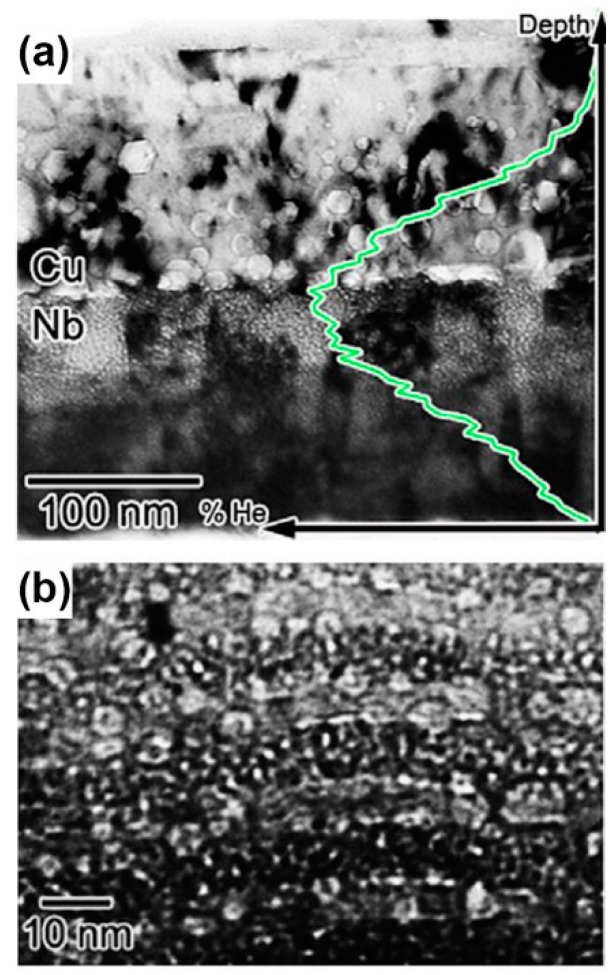
(**a**) Cavities in irradiated Cu/Nb multilayers with 120 nm layer thickness. (**b**) Cavities in He ion irradiated Cu/Nb multilayers with 5 nm layer thickness [46].

**Figure 14 materials-12-02639-f014:**
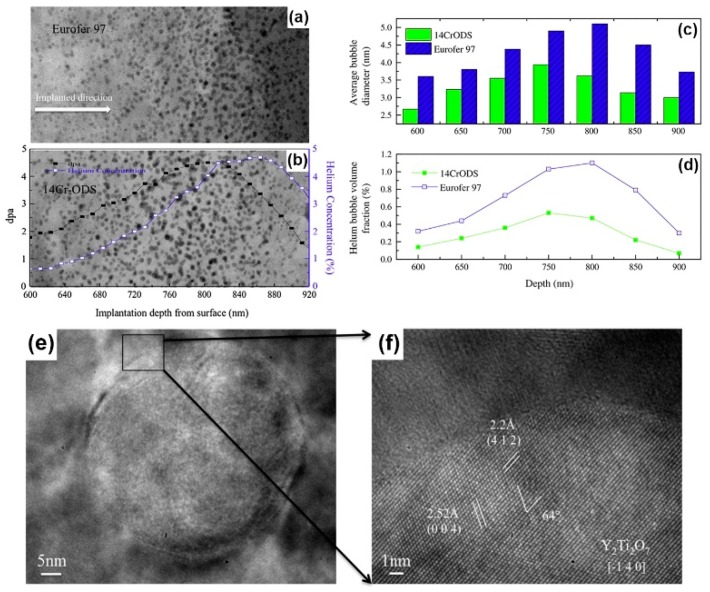
(**a**,**b**) High-angle annular dark field (HAADF) images of the highest helium concentration regions (600–920 nm from the surface) for Eurofer 97 and 14CrODS steel, overlaid with predicted displacement per atom (DPA) and He concentration profile; (**c**,**d**) helium bubble distribution (average bubble diameter and helium bubble volume fraction.) versus depth; (**e**) a Y_2_Ti_2_O_7_ precipitate; (**f**) HRTEM of the pyrochlore structure Y_2_Ti_2_O_7_ [49].

**Figure 15 materials-12-02639-f015:**
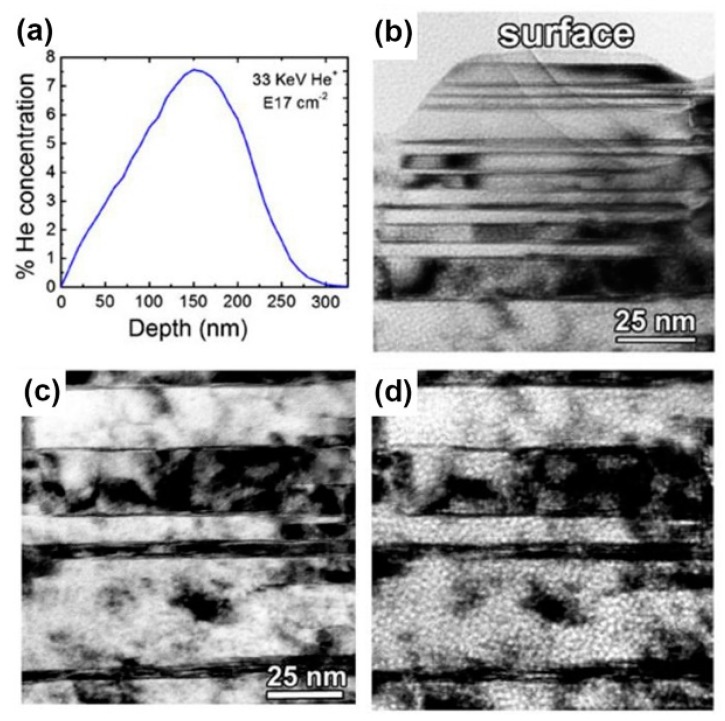
(**a**) Depth profile of He concentration in nanotwined (NT) Cu film implanted to a dose of $1\times 10^{17}\;\mathrm{ions}/{\mathrm{cm}}^{2}$ with 33 keV He ions, calculated using the stopping and range of ions in matter (SRIM). (**b**) Under-focused (−2 um) bright field TEM image from near surface. (**c**,**d**) Near-focus and under-focused (−2 um) bright field TEM images from a depth of 150 nm (peak He concentration) [50].

**Figure 16 materials-12-02639-f016:**
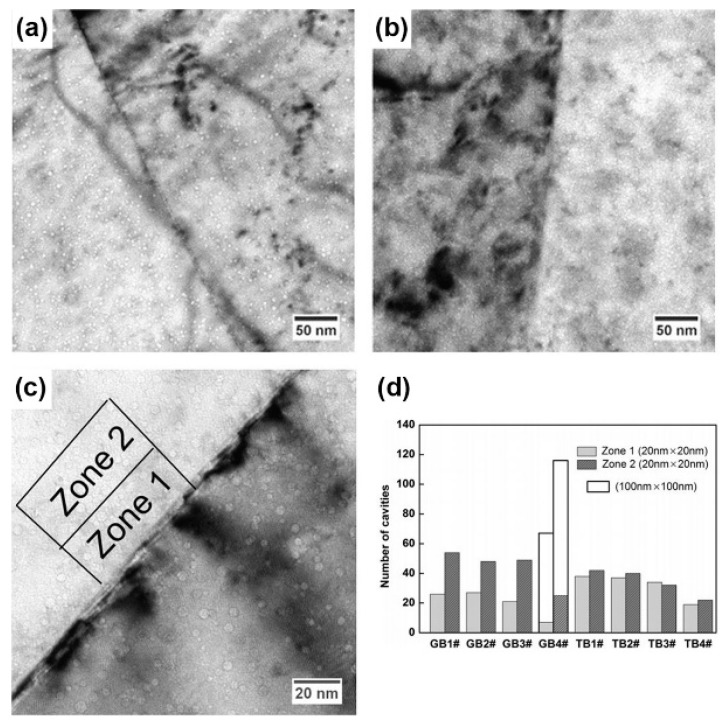
(**a**,**b**) Cavities near general a grain boundary and twin boundary in Cu after He ion irradiation. (**c**) high-magnification micrograph of the cavities near the twin boundary. (**d**) Statistical data of the cavities number adjacent to two types of the boundaries [51].

**Figure 17 materials-12-02639-f017:**
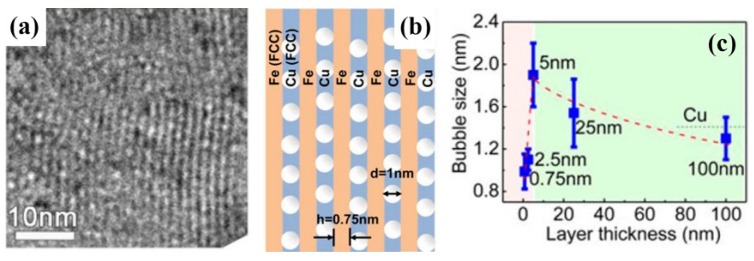
(**a**) TEM images of irradiated fully coherent 0.75 nm Cu/Fe multilayer. (**b**) Schematics illustrate that in a coherent 0.75 nm Cu/Fe multilayer, He bubbles prefer to nucleate in Cu layers and are constricted to reside inside Cu layers. (**c**) The variation of He bubble density with layer thickness h [20].

**Figure 18 materials-12-02639-f018:**
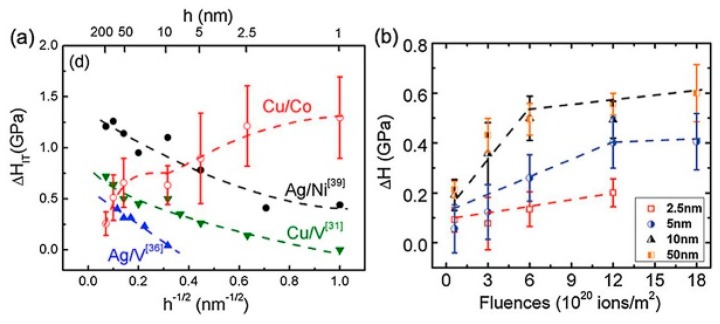
(**a**) Irradiation hardening in He ions irradiated Ag/V, Ag/Ni, Cu/V multilayers scaling with layer thickness. (**b**) Fluence dependence of irradiation hardening in He ion irradiated Cu/V multilayers [39].

**Figure 19 materials-12-02639-f019:**
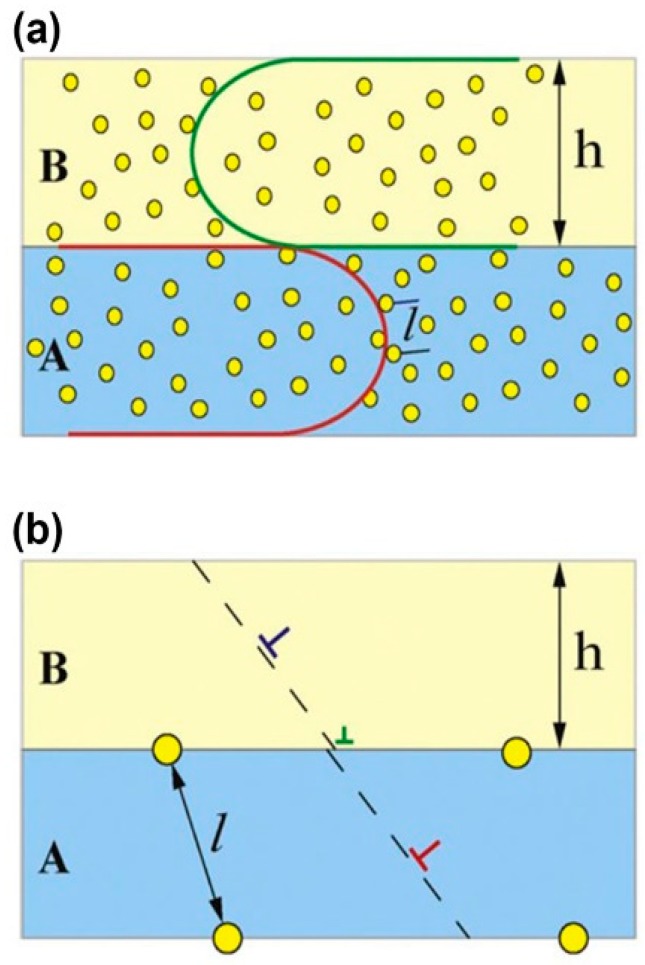
Schematic illustration of the bubble distribution in multilayers (the circles indicate bubbles). (**a**) When the layer thickness is a few tens of nanometers, h > l and the deformation is via confined layer slip. (**b**) When the layer thickness is of the order of a few nanometers, h < l, the yield strength is determined by the crossing of single dislocations across interfaces [43].

**Figure 20 materials-12-02639-f020:**
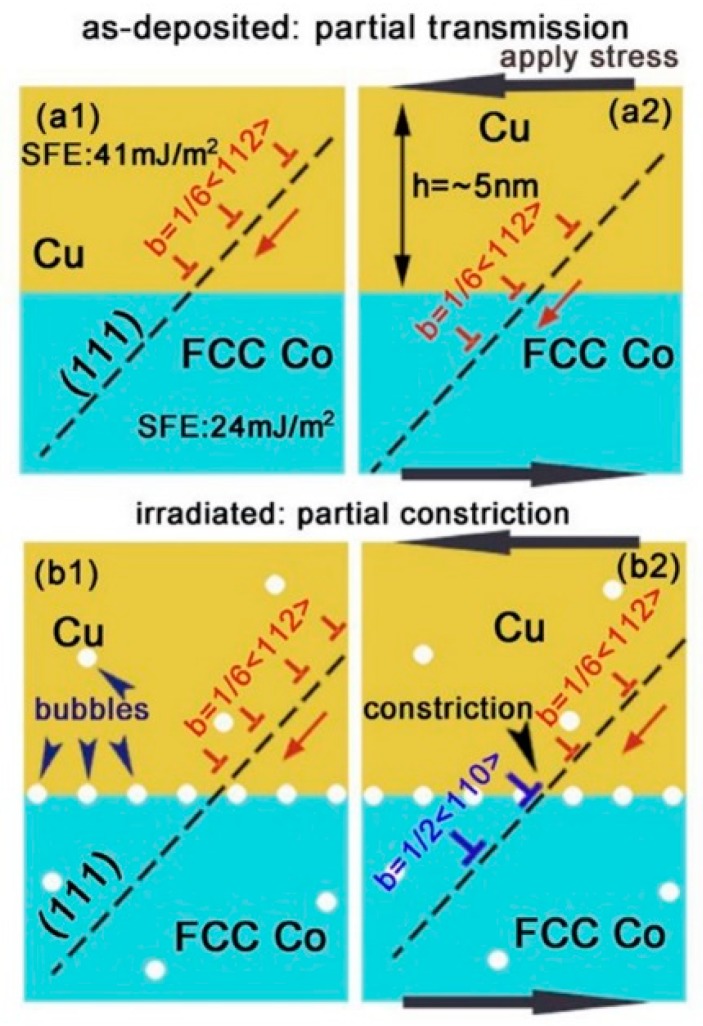
Hypothetical schematics compare strengthening mechanisms in as-deposited and irradiated Cu/Co (100) multilayers at small h (h = 5 nm). (**a1, a2**) In as-deposited films, partials can trespass layer interfaces owing to the low stacking fault energy of Cu and Co. (**b1, b2**) However, after radiation, bubbles at the layer interface disrupt the transmission of partials [56].

**Figure 21 materials-12-02639-f021:**
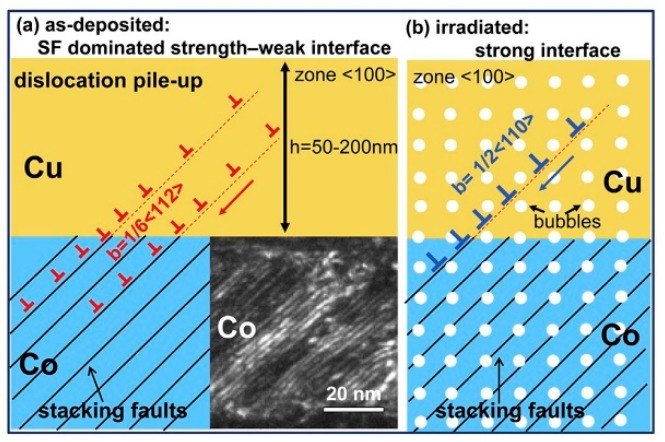
Hypothetical schematics illustrate different strengthening mechanisms in as-deposited and irradiated Cu/Co (100) multilayers at large h (h = 50–200 nm). (**a**) In the as-deposited state, partial dislocations can transmit across the Cu/Co interface relatively easily; (**b**) After radiation, high-density He bubbles are distributed both along the layer interface and within the layers [56].

**Figure 22 materials-12-02639-f022:**
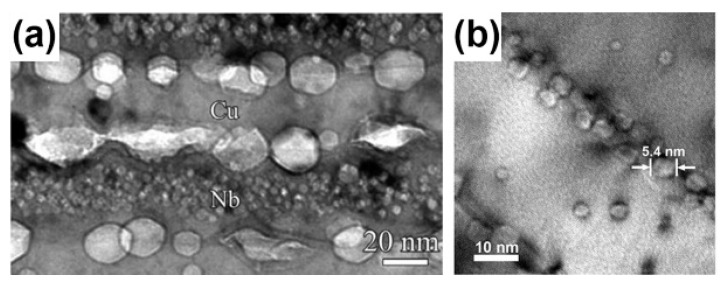
(**a**) Cavities in irradiated Cu/Nb multilayers with 58 nm layer thickness [22]. (**b**) Cavities in irradiated nanocrystalline W [60].

**Table 1 materials-12-02639-t001:** The relationships for multilayers between lattice mismatch, layer thickness, irradiation dose, temperature and cavity size, cavity density.

	Lattice Mismatch	Layer Thickness	Irradiation Dose	Temperature
Cavity size	−	+	+	+
Cavity density	−	+	+	−

“+” means positive relationship, for example, cavity size decreases with layer thickness reducing. “−” means negative relationship, for example, cavity density decreases with lattice mismatch increasing.

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
