# Peer review of "Interface Effects on He Ion Irradiation in Nanostructured Materials"

_materials, 2019, doi:10.3390/ma12162639_

Round 1
Reviewer 1 Report
The paper is a good review of radiation effects of He and the enhancement of materials resistance due to its nanostructure with special attention to multilayers.
Some minor comments/ suggestions to improve the manuscript:
1. The legends and units of axes of grafs in figs 4, 5, 9, 10, 15 and 17 are too small.
2. Abbreviation ARB is not defined.
3. The English is generally good, however, the usage of plural is disturbing at some places. I point out some mistakes:
Lines 241,242,243,248 defects concentration à defect concentration
Line 311 irradiations à irradiation
Line 414 cavities-depleted à cavity-depleted
Line 417 cavities formation à cavity formation
Line 510 cavities spacing à cavity spacing
Line 511 cavities induced hardening à cavity induced hardening
Line 507 50nm < h <200nm
Several places in the text: exam à examination
Provided the authors make the minor corrections, the paper is suitable for publication and no further review is necessary.
Reviewer 2 Report
The authors present their review of interface effects on He ion irradiations in nanostructured materials. They summaries the current understanding of irradiation response in nanostructured materials, including interactions between cavities and interfaces, as well as interface effects on irradiation damage.
In the introduction chapter, the behavior of He in irradiated material under various temperature regimes, the influence of He irradiation on mechanical properties and the way to achieve the irradiation resistance enhancement by introducing interfaces, is presented.
The second chapter summaries the response of He irradiation in nanostructured materials through interaction between cavities and interfaces followed by assessing effects of interfaces on damage caused by irradiation. Irradiation dose and temperature regime were evaluated. The evolution of mechanical properties of irradiated nanostructures was then discussed. Finally, on the bases of the review, three directions of further research in the field were proposed.
This review can be considered as an appropriate summary for the evaluation of research in the given area and the proposal of the direction of research in the following period.
The contribution is written clearly and in good English. I suggest the article for publication.
Author Response
Response: We thank the reviewer very much for his/her appreciation.
Reviewer 3 Report
The authors review much of the literature on He behavior in a variety of conventional and advanced (multilayered) materials. A general description of the mechanisms by which He diffuses and interacts to form clusters is described well, as is the subsequent effect of He-stabilization of cavities and the more pronounced onset of cavity swelling in some dose and temperature regimes. The authors then describe some of the research that has been performed on nanocrystalline and multilayered composites, with a particular focus on the microstructural features and mechanical properties following He ion irradiation.
The manuscript is readable and provides a good summary of prior research on single-ion He implantation into a variety of materials. However, the reviewer believes that the article should be made better through moderate to major revisions prior to acceptance. The encouraged suggestions are (1) to provide a more complete review and (2) to explain what new perspective, insights, etc. that this review provides the scientific community:
(1) Summary but no Synthesis/Analysis: Individual studies are listed sequentially in the text, with some summarizing of the prior author’s conclusions. There are many factors that the authors are taking into consideration: the homologous temperature of the sample during irradiation, the irradiation dose (or fluence) of He atoms implanted, the grain/layer size, and even the misorientation of the interfaces (incoherent, semi-coherent, fully coherent). All of these parameters, and the consequent resulting effects such as hardening contributions, measured volumetric swelling, cavity size, cavity number density, etc. could be put into a summary table and conclusions could easily be drawn by the authors as to the threshold interface characteristics needed to prevent certain micro- and macro-scale effects due to He in the matrix. For example, Zinkle has previously synthesized data from a variety of experiments and has proposed that with a sink strength of 1016m-2, He-stabilized cavity swelling can be suppressed [S.J. Zinkle, L.L. Snead, Designing Radiation Resistance in Materials for Fusion Energy, Annual Review of Materials Research 44(1) (2014) 241-267].
(2) Possible recommendation: From the rate theory approach for sink strength calculation described by Mansur [L. Mansur, Theory and experimental background on dimensional changes in irradiated alloys, Journal of Nuclear Materials 216 (1994) 97-123], the authors could easily calculate the sink strengths in each of the alloys studied (especially for the multilayered composites), which is only a function of grain size, dislocation density, and precipitates (if any). Then the authors could possibly relate all of these studies, and see if the sink strength evaluation approach is a viable way to (1) mitigate hardening, (2) reduce cavity swelling through a refinement of cavity size, etc.
(3) Conventional Materials are discussed, multilayered materials are discussed, but dispersion strengthened materials are not discussed at all, even though Figure 2 mentions the nucleation of He on the surface of precipitates. There are programs in United States, France, and Japan, only to name a few, specifically focusing on the development of dispersion strengthened (oxide, nitride, carbide, etc.), which have become prime candidates for fast and fusion reactor components due to their high strength and irradiation resistance. I would encourage the authors to incorporate at least a few of the plethora of articles published over the last ~20 years or so on nanostructured ferritic alloys, especially since the author uses a figure from Trinkaus’s 2003 review in Figure 2 that shows how He nucleates along a precipitate boundary already, with no further discussion about precipitate interfaces being used as trapping sites for He.
(4) The authors make an interesting comment at the end of section 2.2.3 when discussing coherent boundaries: “However, recent studies show that Kr ion irradiation could produce abundant steps (containing dislocations) [44, 45] on the CTBs and thus improve the ability for absorbing point defects of CTBs [45].” In most of the cited works in this review article, the authors limit their discussion on interface/He interactions to studies which simply use only He ion irradiation/implantation. However, the point of the article, as elucidated from the first sentence of the abstract, mentions “In advanced fission and fusion reactors, structural materials suffered from high dose irradiation by energetic particles would subject to severe microstructure damage”. Although a variety of dual-ion irradiation studies exist in the literature probing the effect of irradiation damage and He implantation simultaneously, with [C.M. Parish, K.A. Unocic, L. Tan, S.J. Zinkle, S. Kondo, L.L. Snead, D.T. Hoelzer, Y. Katoh, Helium sequestration at nanoparticle-matrix interfaces in helium + heavy ion irradiated nanostructured ferritic alloys, Journal of Nuclear Materials 483 (2017) 21-34] being only one example, the review feels somewhat incomplete from a nuclear perspective. Since the authors acknowledge that irradiation cascades due to neutrons or larger ions (like Kr) can alter interface structures, I am surprised that this discussion is not attempted in this review.
(5) The author does a good job keeping with their operational definitions of bubbles and voids within the review article, saying that a void is any cavity that is under-pressurized, while any bubble is a cavity that is over-pressurized. For example, the author described well the differences between larger faceted voids in Cu and smaller spherical bubbles in Nb in Section 2.2.2. However, since a clear delineation has been bade between voids and bubbles, it leaves what appear to be gaps in the discussion in some parts of the review. For instance, section 2.1 is titled “The interactions between cavitiesand interfaces, while the two subsections within it are titled “2.1.1 Void-denuded zones near interfaces” and “2.1.2Voids adhere to interfaces or cross interfaces”. If two subsections are dedicated specifically to the under-pressurized cavities, why is there no discussion about the distributions of over-pressurized cavities within materials with respect to interfaces. In other words, if the parent section mentions that cavities (bubbles and voids) are going to be discussed, it is unclear why no discussion of bubbles is had in the subsections that follow.
(6) The reviewer would highly recommend that the authors have a technical reviewer look over the grammar of the article. The sentences are coherent and understandable, but improvements can be made to the English grammar used.
Round 2
Reviewer 3 Report
The authors' additions strengthen the paper significantly. I believe that the manuscript can be accepted in its present form.